# ENO1 from *Mycoplasma bovis* Disrupts Host Glycolysis and Inflammation by Binding ACTB

**DOI:** 10.3390/biom15081107

**Published:** 2025-08-01

**Authors:** Rui-Rui Li, Xiao-Jiao Yu, Jia-Yin Liang, Jin-Liang Sheng, Hui Zhang, Chuang-Fu Chen, Zhong-Chen Ma, Yong Wang

**Affiliations:** College of Animal Science and Technology, Shihezi University, Shihezi 832000, China; 20242013058@stu.shzu.edu (R.-R.L.); yuxiaojiao@stu.shzu.edu.cn (X.-J.Y.); liangjiayin@stu.shzu.edu.cn (J.-Y.L.); wangwenjia@stu.shzu.edu.cn (J.-L.S.); zhangqianyi@stu.shzu.edu.cn (H.Z.); ccf@shzu.edu.cn (C.-F.C.)

**Keywords:** α-enolase, β-actin, interaction, glycolysis, metabolic–inflammatory cascade

## Abstract

*Mycoplasma bovis* is an important pathogen that is associated with respiratory diseases, mastitis, and arthritis in cattle, leading to significant economic losses in the global cattle industry. Most notably in this study, we pioneer the discovery that its secreted effector ENO1 (α-enolase) directly targets host cytoskeletal proteins for metabolic–immune regulation. Using an innovative GST pull-down/mass spectrometry approach, we made the seminal discovery of β-actin (ACTB) as the primary host target of ENO1—the first reported bacterial effector–cytoskeleton interaction mediating metabolic reprogramming. ENO1–ACTB binding depends on a hydrogen bond network involving ACTB’s 117Glu and 372Arg residues. This interaction triggers (1) glycolytic activation via Glut1 upregulation, establishing Warburg effect characteristics (lactic acid accumulation/ATP inhibition), and (2) ROS-mediated activation of dual inflammatory axes (HIF-1α/IL-1β and IL-6/TNF-α). This work establishes three groundbreaking concepts: (1) the first evidence of a pathogen effector hijacking host ACTB for metabolic manipulation, (2) a novel ‘glycolysis–ACTB–ROS-inflammation’ axis, and (3) the first demonstration of bacterial proteins coordinating a Warburg effect with cytokine storms. These findings provide new targets for anti-infection therapies against *Mycoplasma bovis*.

## 1. Introduction

*Mycoplasma bovis* (*M. bovis*) is a significant pathogen affecting the cattle industry, capable of causing various diseases including bovine respiratory disease complex, mastitis, and arthritis, leading to substantial economic losses in the global cattle industry [1,2,3,4]. This pathogenic microorganism can co-infect with multiple pathogens, such as *Pasteurella multocida* [5], *Histophilus somni* [6], bovine respiratory syncytial virus [7], bovine herpesvirus 1 [8], bovine viral diarrhea virus [6], *M. bovis* [9], and type 3 parainfluenza virus [7], exacerbating these diseases in infected animals. Furthermore, *M. bovis* is one of the primary pathogens of the bovine respiratory disease complex [10,11]. As a small prokaryote lacking a cell wall, *M. bovis* interacts with host cells through surface proteins, among which ENO1 (α-enolase) has gained attention due to its multifunctionality [12].

ENO1 is an evolutionarily highly conserved glycolytic enzyme that has been confirmed to have non-classical pathogenic functions in various pathogens. Studies have found that the surface ENO1 of *M. bovis* can promote pathogen adhesion to host cells by binding to plasminogen (Plg) [12]. Notably, in addition to its adhesive function, ENO1 has also been shown to influence pathogen survival and pathogenicity by regulating the glycolytic pathway in other pathogens, such as *Candida albicans* [13]. As an archaea lacking oxidative phosphorylation pathways, *M. bovis* relies exclusively on glycolysis for ATP generation under both normoxic and hypoxic conditions [14]. Given that the survival of mycoplasma is highly dependent on glycolytic metabolism [15], we speculate that ENO1 may play a broader regulatory role in the pathogenic process of *M. bovis*.

The glycolysis–HIF-1α–inflammation signaling axis plays a critical role in pathogen infection. Under hypoxic conditions, the stability of HIF-1α is increased, leading to the upregulation of glycolysis-related enzyme expression [16]. Abnormal glycolysis can result in the accumulation of reactive oxygen species (ROS) [17]. ROS can both stabilize HIF-1α [18] and activate the NLRP3 inflammasome, promoting the release of inflammatory factors such as IL-1β [19,20], thus forming a positive feedback loop of ‘metabolism–oxidative stress–inflammation’. For example, in sepsis, LPS-activated macrophages undergo glycolytic reprogramming via mTOR/HIF-1α/PFKFB3 signaling, driving ROS-dependent NLRP3 inflammasome activation and IL-1β release [21]. However, it remains unclear whether *M. bovis* ENO1 regulates the host pathological process through this pathway.

While *M. bovis* ENO1 is known to promote adhesion via Plg binding, its potential role in modulating host metabolic and inflammatory pathways central to pathogenesis remains unexplored. Based on the importance of the glycolysis–HIF-1α–inflammation axis in infection and *M. bovis*’s reliance on glycolysis, this study specifically aimed to (1) Identify and characterize the interaction between *M. bovis* ENO1 and host cell proteins, focusing on ACTB and its critical binding sites. (2) Elucidate whether and how the ENO1–ACTB interaction induces dysregulation of host glycolysis, ROS production, HIF-1α signaling, and IL-1β-mediated inflammation. (3) Assess the functional significance of this interaction and pathway dysregulation for *M. bovis* persistence within host cells.

## 2. Materials and Methods

### 2.1. Cells and Strains

The *M. bovis* strain XJ-01 was isolated and preserved in our laboratory. Embryonic bovine lung (EBL) cells were generously provided by Professor Guo Aizhen’s team from Huazhong Agricultural University. The HEK293T cells were purchased from the Cell Resource Center, the Institute of Basic Medical Sciences of the Chinese Academy of Medical Sciences/Peking Union Medical College (Beijing, China). Both EBL and HEK293T cells were cultured in Dulbecco’s modified Eagle medium (DMEM) (11995, Solarbio, Beijing, China) supplemented with 10% fetal bovine serum (FBS) (16000-044, Gibco, CA, USA) under 5% CO_2_ conditions.

### 2.2. Construction of Plasmids and Interfering Fragments

The ENO1 open reading frame (ORF) sequence of the *M. bovis* PG45 strain (WP_013456550.1) was obtained from the NCBI database. The ENO1 ORF was synthesized by Sangon Biotech. A Myc tag was fused at the C-terminus of the ENO1 ORF and cloned into the pcDNA3.1 (Invitrogen, Carlsbad, CA, USA) vector, resulting in the construction of the pcDNA3.1-Ats-1 recombinant vector. The ENO1 ORF was also ligated to the pET-22b (+) vector, with a GST tag sequence added at the C-terminus, leading to the construction of the pET-22b-ENO1-GST recombinant vector. This vector was transformed into Escherichia coli DE3 to establish the expression strain for ENO1-GST.

Five potential interacting proteins’ (VIM NC_037340.1, ACTB NC_037352.1, KRT10 NC_037346.1, KRT14 NC_037346.1, and ALB NC_037333.1) ORF sequences were obtained from the NCBI database. An HA tag was fused to the C-terminus of each ORF and connected to the pcDNA3.1 (Invitrogen, USA) vector. The recombinant plasmids were extracted using a large-scale plasmid extraction kit (DP120-01, Tiangen, Beijing, China) for transfection purposes.

The transcript of ACTB (NM_173979.3) was retrieved from the NCBI database. Three small interfering RNA (siRNA) fragments targeting the ACTB gene were designed using online software from Thermo Fisher (https://rnaidesigner.thermofisher.com/rnaiexpress/setOption.do?designO) (accessed on 1 May 2024). The specific sequences can be found in Appendix A.

### 2.3. Transfection of Plasmids and Infection with M. bovis

Cells were prepared when they reached a density of 70–80%. Recombinant plasmids and interference fragments were transfected into the cells using jetPRIME^®^ transfection reagent (101000046, Polyplus, Illkirch, France) according to the manufacturer’s instructions.

EBL cells were grown to the logarithmic phase and passaged to 6 cm cell culture plates after digestion with trypsin, followed by overnight incubation. The culture of *M. bovis* XJ-01 was centrifuged three days post-culture, washed twice with PBS, and subjected to color change unit (CCU) counting for *M. bovis*. The CCU assay was performed by serial 10-fold dilutions of mycoplasma cultures in phenol red-containing broth medium, incubated at 37 °C for 7–14 days, with the highest dilution showing color change defined as 1 CCU/mL. Subsequently, *M. bovis* was added to the culture plates at a multiplicity of infection (MOI) of 1000. The cells were infected for 12 h at 37 °C in a 5% CO_2_ atmosphere.

### 2.4. Detection of Glucose, Lactic Acid, ATP, and ROS Levels

After processing the cells from each group, glucose levels were measured using a glucose detection kit (S0201S, Beyotime, Beijing, China), L-lactic acid (L-LA) levels were assessed using an L-lactic acid detection kit (BC2235, Solarbio, China), ATP levels were determined using an ATP detection kit (S0026, Beyotime, China), and ROS levels were evaluated using an ROS detection kit (S0033S, Beyotime, China) according to the manufacturer’s instructions.

### 2.5. Western Blot

This study employs Western blotting in conjunction with ImageJ software (V1.8.0.112) to analyze the expression levels of ENO1 over different time periods and the protein expression levels of Glut1, ACTB, HIF-1α, and IL-1β in various sample groups. Cells subjected to different treatments were lysed using Native Lysis Buffer (R0030, Solarbio, China) and protease inhibitors (P6730, Solarbio, China) to extract total cellular proteins. The protein concentration of the samples was determined using a BCA protein quantification kit (BCA1-1KT, Sigma, MO, USA). After adding SDS-PAGE Loading Buffer (CW0027, CWBIO, Beijing, China) to the protein extracts, samples were boiled at 100 °C for 10 min before undergoing SDS-PAGE. A constant current of 230 mA was applied for membrane transfer, and the PVDF membrane was blocked with 5% non-fat milk at room temperature for 2 h, followed by incubation with primary and secondary antibodies for 2 h each. A SuperSignal West Femto Trial Kit (34094, Thermo Fisher Scientific, Waltham, MA, USA) was utilized for color development. Semi-quantitative analysis of the Western blot results was performed using ImageJ software. Details of the primary and secondary antibodies are provided in Appendix A.

### 2.6. Immunofluorescence Experiment

This study employs immunofluorescence experiments to verify the expression of the Glut1 protein and the interactions between ENO1 and ACTB. Cell slides from each group were fixed with 4% paraformaldehyde at room temperature for 20 min, followed by three washes with PBS. Cells were permeabilized with 0.1% Triton X-100 for 10 min and washed three times with PBS. A 5% BSA blocking solution was applied at room temperature for 30 min, followed by one wash with PBS. Glut1 antibody or a combination of anti-Myc tag antibody (labeling ENO1) and anti-HA tag antibody (labeling ACTB) was added, and the samples were incubated overnight at 4 °C, followed by three washes with PBS. Corresponding species-specific fluorescent secondary antibodies were added and incubated in the dark at 37 °C for 1 h, followed by three washes with PBS. DAPI staining solution was incubated in the dark for 3 min, followed by three washes with PBS. An anti-fluorescence quenching mounting medium was applied and coverslips were mounted and stored in the dark until imaging. Images were collected and multi-channel images were saved. Details of the primary antibodies and fluorescent secondary antibodies are provided in Appendix A.

### 2.7. GST Pull-Down

A total of 0.5 mL of GST beads (C650031-0010, Sangon Biotech, Shanghai, China) was taken, and the supernatant was discarded after magnetic separation. An equal volume of binding buffer was added and mixed thoroughly, followed by two washes. The ENO1-GST protein was mixed with the beads and incubated at room temperature with inversion for 30 min, after which the supernatant was discarded (the flow-through was retained). Two times the volume of wash buffer was added to wash the beads, repeating this step twice. Subsequently, EBL lysis buffer was added and incubated for 30 min, followed by magnetic separation to retain the supernatant. An equal volume of elution buffer was added, and the mixture was incubated at room temperature for 10 min before magnetic separation to collect the elution liquid. The elution liquid was subjected to SDS-PAGE, and after silver staining, the target band was excised for qualitative analysis by mass spectrometry (Beijing Novogene Co., Ltd., Beijing, China).

### 2.8. Co-Immunoprecipitation (Co-IP)

This method was employed to detect ENO1-interacting proteins. Potential interacting protein genes identified by mass spectrometry qualitative analysis were C-terminally fused with an HA tag and cloned into the pcDNA3.1 vector (Invitrogen, USA). These constructs were co-transfected with pcDNA3.1-ENO1 (Myc-tagged) into HEK293T cells. After 48 h of culture, total cellular proteins were extracted using RIPA buffer (R0020, Solarbio, China) supplemented with protease inhibitors (P6730, Solarbio, China). Rabbit anti-HA tag antibody (ab236632, Abcam, Waltham, MA, USA) or rabbit anti-Myc tag antibody (ab9106, Abcam, USA) was added to the lysates, followed by incubation at 4 °C with gentle agitation (120 rpm) for 3 h. Protein G-Agarose beads (P7700, Roche, Mannheim, Germany) were then added, and the mixture was incubated overnight at 4 °C with agitation (120 rpm). The next day, immunocomplexes were pelleted by centrifugation and washed three times with PBS (P1020, Solarbio, China). Precipitates were boiled in SDS-PAGE loading buffer (CW0027, CWBIO, China) for 10 min and subjected to immunoblotting analysis. Primary antibodies used were mouse anti-Myc tag antibody (ab32, Abcam, USA) or mouse anti-HA tag antibody (ab18181, Abcam, USA), with goat anti-mouse IgG H&L (ab6789, Abcam, USA) as the secondary antibody. Protein bands were visualized using the SuperSignal™ West Femto Maximum Sensitivity Substrate Kit (34094, Thermo Fisher Scientific, USA).

### 2.9. Key Interaction Amino Acid Prediction

The structures of the target proteins ENO1 and ACTB were modeled using Swiss-Model homology modeling software based on their amino acid sequences (https://swissmodel.expasy.org/) (accessed on 1 March 2024). Using the HDOCK software (v.1.1.0.), each protein was set as rigid, with the docking contact points configured to the entire surface. A total of 100 conformations were generated post-docking, and the docking scores were calculated based on the knowledge-based iterative scoring function ITScorePP. A more negative docking score indicates a more likely binding model. The conformations with the most negative scores were selected using the scoring function and optimized using the Minimization module within the MOE 2019.1 software platform. The optimization results were visualized and analyzed using the Pymol2.1 software. The amino acids most likely to be the binding sites for ACTB were selected to create different mutant docking models and docking scores were calculated. Based on the amino acid sequences of the mutant schemes with the highest scores, three ACTB mutant plasmids were synthesized.

### 2.10. RNA Extraction and RT-qPCR

Total RNA from cells was extracted using an Ultrapure RNA Kit (CW0581M, CWBIO, Beijing, China) and reverse transcription was performed with the HiFiScript cDNA Synthesis Kit (CW2569M, CWBIO, Beijing, China). Quantitative detection was conducted using the SYBR green dye method. The cycling conditions were set at 95 °C for 5 s for denaturation, 95 °C for 5 s for extension, and 60 °C for 30 s for annealing, with a total of 45 cycles. Samples were run on the QuantStudio 5 real-time PCR system (Thermo Fisher Scientific, MA, USA). Data were normalized based on the expression levels of GAPDH in each sample. All experiments were conducted in triplicate. The RT-qPCR primers for IL-6 and TNF-α are listed in Appendix A.

### 2.11. Counting of Intracellular M. bovis

After 12 h of *M. bovis* infection, the culture medium was discarded, and the cells were washed twice with PBS. Triton X-100 (20107ES20, Yeasen, Shanghai, China) was added to lyse the cells and release the intracellular *M. bovis*. A 20 μL aliquot of the cell suspension was placed on solid culture plates and incubated for 7 days. The growth of the colonies was then observed and photographed. The relative optical density of the colonies was analyzed using ImageJ software to quantify the number of *M. bovis* in the cells.

### 2.12. Quantitative and Statistical Analysis

All data presented herein represent the results from three separate experiments and are mean ± SD. GraphPad Prism software (8.3.1) was used to draw the figures in this paper. The SPSS software (SPSS statistics 25.0) was used for data statistical analysis. Immunoblotting results were semi-quantified using the ImageJ software. Data were compared with different groups using one-way analysis of variance (ANOVA), Student–Newman–Keuls (SNK) tests, and Student’s *t*-tests. Ns—not significant (*p* > 0.05), 0.01 < ** p* < 0.05, *** p* < 0.01, **** p* < 0.001, and ***** p* < 0.0001.

## 3. Result

### 3.1. ENO1 and M. bovis Infection Induce Glycolysis in Host Cells

By constructing a eukaryotic expression vector for ENO1 and successfully transfecting EBL cells (Appendix A), Western blot analysis confirmed that the ENO1 protein was stably and efficiently expressed in host cells 48 h post-transfection, providing a reliable experimental model for subsequent metabolic studies. Using commercial kits, we systematically detected the expression of ENO1 and the host cells infected by *M. bovis*. The results showed that, compared to the control group, both the ENO1 expression group and the *M. bovis* infection group exhibited a significant increase in intracellular glucose concentration (Figure 1A). The lactate production also significantly increased (Figure 1B). Quantitative analysis with fluorescent labeling indicated an upregulation of Glut1 protein expression (Figure 1C,D), and Western blot results further confirmed the increase in Glut1 expression (Figure 1E,F). The upregulation of Glut1 led to an enhanced glucose transport capacity across the cell membrane; the accumulation of lactate indicates a significant activation of the glycolytic pathway. These results suggest that *M. bovis* induces metabolic reprogramming of host cells through the ENO1 effector protein, characterized by enhanced glucose uptake and promotion of glycolysis. This metabolic regulation may be an important mechanism for *M. bovis* to establish persistent infections within the host.

### 3.2. Interaction of ENO1 with Host Cell ACTB

To further investigate the interaction network of ENO1 (GST) protein, this study employed a magnetic bead pull-down combined with mass spectrometry analysis to systematically screen its interacting proteins. A prokaryotic expression system was used to successfully induce the expression of ENO1 (GST) fusion protein. SDS-PAGE analysis revealed (Appendix A) that the protein purified through affinity chromatography was highly enriched in the elution buffer, with a single band observed in the electrophoresis, indicating the acquisition of high-purity target protein. Through GST pull-down experiments combined with SDS-PAGE analysis (Figure 2A), specific protein bands were observed in the experimental group, suggesting the presence of potential interacting proteins that bind to ENO1 (GST). Mass spectrometry analysis and rigorous screening successfully identified five candidate interacting proteins (Figure 2B). Co-IP experiments confirmed (Figure 2C) that interaction signals could only be detected in the precipitate when ENO1 was co-expressed with ACTB. Reverse pull-down experiments further validated this specific binding. Fluorescence co-localization experiments demonstrated that ENO1 and ACTB exhibited spatial co-localization within the cells (Figure 2D), providing evidence from multiple perspectives for the interaction between the two. This study systematically screened and verified ACTB as an important interacting protein of ENO1, providing new experimental evidence to elucidate the biological function of ENO1.

### 3.3. ENO1 Regulates Glycolytic Reprogramming by Modulating ACTB

To elucidate the molecular mechanism of ACTB in the regulation of the glycolytic pathway by ENO1, this study first employed Western blot technology to systematically analyze the effects of ENO1 overexpression and *M. bovis* infection on the expression of the ACTB protein in host cells. Experimental data showed that, compared to the control group, the level of ACTB protein in host cells was significantly upregulated after 48 h of continuous ENO1 expression; similarly, *M. bovis* infection for 12 h also induced an increase in ACTB expression (Figure 3A). Quantitative analysis of protein bands conducted using ImageJ software further confirmed this regulatory effect (Figure 3B). These results collectively indicate that the *M. bovis* effector protein ENO1 has a positive regulatory function on the expression of ACTB in host cells, suggesting that ACTB may serve as a key effector molecule in the remodeling process of the ENO1-mediated glycolytic pathway.

Three specific siRNA interference fragments targeting the ACTB gene (siRNA ACTB 328/961/1021) were designed. Total RNA was extracted after transfecting EBL cells for 12 h, followed by RT-qPCR detection. The results showed (Appendix A) that the gene silencing efficiency of the siRNA ACTB 961 treatment group reached 71.7%, significantly outperforming the other interference fragments compared to the blank control group that was not transfected with siRNA. Based on these results, siRNA ACTB 961 was selected for subsequent functional studies of the ACTB gene.

To elucidate the role of ACTB in the regulation of the glucose metabolism pathway mediated by ENO1, we systematically assessed the energy metabolism characteristics of cells from different treatment groups (Figure 3C–E). The results indicated that in the ENO1 overexpression group, the intracellular glucose uptake increased and lactic acid production was elevated, while ATP levels decreased. Conversely, in the siRNA–ACTB treatment group an opposite metabolic phenotype was observed, with reductions in glucose and lactic acid levels and a restoration of ATP production. In the siRNA–ACTB and ENO1 co-transfection group, no significant changes were observed in glucose, lactic acid, and ATP levels. These findings suggest that ENO1 promotes glycolytic flux by upregulating ACTB expression, and the absence of ACTB can reverse the Warburg effect induced by ENO1. The metabolic regulatory function of ENO1 is strictly dependent on the expression of ACTB. This research suggests the central role of the ENO1–ACTB axis in metabolic reprogramming, providing new insights into the molecular mechanisms by which pathogens regulate host energy metabolism.

### 3.4. ENO1 Promotes the Inflammatory Response of Host Cells by Regulating ACTB

The study indicated that the accumulation of lactate/succinate due to glycolytic metabolic dysregulation stabilized HIF-1α protein by inhibiting PHD activity, thereby promoting its nuclear translocation and driving IL-1β transcriptional activation. This, in turn, enhanced glycolytic metabolism through a positive feedback mechanism, forming a self–regulating network termed the ‘glycolysis-HIF-1α–IL-1β’ axis. Western blot analysis showed that ENO1 overexpression significantly upregulated the expression of HIF-1α and IL-1β proteins, while ACTB gene silencing effectively blocked this effect (Figure 3F). This revealed that the ENO1–ACTB interaction influenced the activity of the HIF-1α/IL-1β signaling axis by regulating the balance between glycolytic and mitochondrial metabolism. Further studies demonstrated that this molecular interaction exhibited multi-target characteristics in regulating the pro-inflammatory factor network. RT-qPCR analysis confirmed that ENO1 overexpression simultaneously activated IL-6 and TNF-α transcription, but this effect was significantly diminished in the ACTB functional inhibition group (Figure 3G,H). Collectively, these findings systematically elucidated how ENO1 amplified the inflammatory response through synergistic action with ACTB, employing a dual mechanism of metabolic reprogramming and transcriptional regulation. This work provided a molecular basis for therapeutic strategies targeting the metabolic–inflammatory coupling node.

Based on the phenomenon of ROS accumulation caused by the abnormal activation of glycolysis leading to the inhibition of mitochondrial oxidative phosphorylation and dysfunction of the electron transport chain (ETC), this study further explores the regulatory role of the ENO1–ACTB interaction on intracellular redox homeostasis. Using fluorescence probe detection technology, we found that the expression of ENO1 significantly enhanced the intensity of ROS fluorescence signals, and its dynamic changes were positively correlated with the activation process of glycolysis. Conversely, silencing ACTB expression through siRNA interference resulted in a noticeable decline in ROS levels (Figure 3I,J). These results reveal that the interaction between ENO1 and ACTB can drive glycolytic metabolic reprogramming, inducing an explosive generation of ROS associated with mitochondrial dysfunction, thereby establishing a cascade regulatory mechanism of “glycolytic activation–ROS accumulation–oxidative stress enhancement.”

### 3.5. The 117Glu and 372Arg Residues of ACTB Are Critical Interaction Amino Acid Sites

Molecular docking analysis using HDOCK revealed a significant complementarity in the spatial structure between ENO1 and ACTB proteins (Figure 4A). The two proteins form a stable complex through multiple modes of intermolecular interactions. A hydrogen bond network (e.g., Ser-372/Arg-372, Asp-376/Ser-368), salt bridge interactions (e.g., Lys-383/Glu-117), and hydrophobic interactions (e.g., Tyr-137/Phe-375) collectively maintain the stability of the complex. The validity of this model was confirmed by binding free energy calculations, identifying key residues such as 117Glu and 372Arg in ACTB.

To further validate the role of key amino acid residues in the ACTB protein in the formation of the ENO1–ACTB protein complex, we performed point mutations on the critical residues identified in the molecular docking analysis. Specifically, we mutated key sites such as 117Glu and 372Arg in the ACTB protein (Figure 4B) and constructed mutant plasmids pcDNA3.1-mut ACTB II, pcDNA3.1-mut ACTB III, and pcDNA3.1-mut ACTB IV. Through CO-IP experiments, we found that mut ACTB II significantly weakened its binding capacity to ENO1 (Figure 4C,D), while mut ACTB III/IV exhibited enhanced binding, indicating that these two sites may indirectly regulate protein interactions through an allosteric effect. These findings not only clarify the critical roles of the 117Glu and 372Arg residues in the ENO1–ACTB interaction but also reveal the complexity of the dynamic regulation of this complex.

### 3.6. M. bovis Infection Regulates Host Cell Glycolysis Through ACTB

To further elucidate the biological significance of the ENO1–ACTB interaction during pathogen infection, we examined the changes in glucose, lactate, and ATP concentrations in cells following *M. bovis* infection. Metabolic analysis revealed that *M. bovis* XJ01 infection significantly altered the energy metabolism characteristics of host cells, characterized by increased glucose levels and lactate production alongside a decrease in ATP levels (Figure 4E–G). Notably, when ACTB expression was altered through siRNA interference or site-directed mutagenesis, the metabolic reprogramming induced by pathogen infection was significantly reversed. Glucose concentrations and lactate production decreased, while ATP synthesis capacity was restored (Figure 4E–G). This research suggests that ACTB, as a key regulatory factor, may play a central role in maintaining the dynamic balance of intracellular energy metabolism by regulating cytoskeletal reorganization or metabolic enzyme activity during the glycolytic activation and mitochondrial function inhibition induced by *M. bovis* infection.

### 3.7. ACTB-Mediated Regulation of Inflammatory Response and Intracellular Survival of M. bovis

The results indicated that infection with *M. bovis* XJ01 comprehensively activated the host cell stress response network. Western blot results showed that infection significantly upregulated the expression of HIF-1α and IL-1β (Figure 5A), suggesting activation of hypoxic stress and the NF-κB pathway. RT-qPCR analysis confirmed that the transcription levels of IL-6 and TNF-α were simultaneously elevated (Figure 5B,C), and these effects depended on ACTB-mediated cytoskeletal reorganization and signal transduction. DCFH-DA assays revealed a significant increase in ROS levels (Figure 5D,E), which, together with metabolic reprogramming features (activation of glycolysis/ATP inhibition), constituted an ACTB-dependent ‘metabolic–oxidative stress–inflammatory’ cascade response. Gene silencing and functional loss mutation experiments confirmed that ACTB acted as a molecular hub, coordinating dynamic changes in the cytoskeleton while regulating the Rho/NF-κB signaling axis. Simultaneously, it drove metabolic abnormalities, oxidative stress, and inflammatory responses, thereby forming a positive feedback regulatory loop.

To elucidate the regulatory role of the ENO1–ACTB interaction on the intracellular survival capacity of *M. bovis*, we further analyzed the correlation between the expression level of host cell ACTB and the pathogen load. The relative gray value analysis of plaques indicated that ACTB overexpression significantly reduced the intracellular pathogen load, whereas ACTB gene silencing or loss-of-function mutations led to a marked enhancement in pathogen proliferation (Figure 5F,G). Further mechanistic analysis revealed that ENO1 targets and induces ACTB expression, triggering a metabolic reprogramming mechanism characterized by the inhibition of oxidative phosphorylation and activation of glycolysis in host cells, thereby establishing a stress microenvironment characterized by ROS accumulation and pro-inflammatory factor network activation. This ENO1–ACTB axis-driven metabolic–inflammatory coupling response system may play a crucial role in regulating the balance between host defense and pathogen survival, providing new insights into the molecular mechanisms of intracellular pathogen–host metabolic interactions.

## 4. Discussion

*Mycoplasma* infections cause significant economic losses in global livestock, with *M. bovis* being a challenging target for prevention and control due to its multiple drug resistance and persistent infection characteristics [5]. Recent studies have shown that the phenomenon of pathogen effector proteins hijacking host metabolic pathways to establish infections is widespread. For instance, *Toxoplasma gondii* manipulates the signaling pathways of host cells through its secreted effector molecules to promote its survival and reproduction within host cells [22]. Similarly, *Leishmania infantum* regulates the mitochondrial metabolism of host macrophages by hijacking the SIRT1–AMPK axis of host cells, thereby supporting its growth within the host [23]. This study systematically reveals the molecular mechanism by which the *M. bovis* effector protein ENO1 targets the host cytoskeletal protein ACTB to regulate the metabolic–inflammation network (Figure 6). Through multi-omics techniques, we have identified the key binding site (117Glu/372Arg) of the ENO1–ACTB interaction for the first time and confirmed that this interaction triggers the Warburg effect by inducing an imbalance between glycolysis and oxidative phosphorylation (manifested as increased glucose uptake, lactate accumulation, and ATP inhibition), thereby activating the HIF-1α/IL-1β and IL-6/TNF-α dual signaling axes. This finding not only expands our understanding of the pathogenic mechanisms of *M. bovis* but also provides new insights into the evolutionary strategies of pathogens hijacking the host metabolism.

*M. bovis* mediates host infection through various surface adhesins, including well-defined host targets such as NOX [24], MbfN [25], FBA [24], FBA [25], FBA [26,27], TrmFO [28], α-enolase [12], MilA [29], P27 [30], LppA [31] and LppB [32], as well as functions that are not yet fully elucidated, such as VpmaX [33], P26 [34], VSP [35,36], Mbov-0503 [37], and the 24 kDa protein [38]. Among these virulence factors, ENO1 has emerged as a core effector molecule in the pathogenic process due to its unique ‘metabolic–adhesion dual function’. As a key surface-exposed virulence factor, the C-terminal domain of ENO1 (amino acids 250–434) specifically binds to host Plg, directly mediating the adhesion and invasion of the pathogen to host cells [12,39]. Additionally, we reveal for the first time the interaction between secreted ENO1 and ACTB, which induces abnormal host glycolysis. This dual-function mechanism bears similarity to *Candida albicans* ENO1, which also exhibits metabolic and adhesive roles [13]. However, unlike *Candida albicans* ENO1, *M. bovis* ENO1 demonstrates a more pronounced and distinct immunomodulatory characteristic; specifically, it enhances IL-1β secretion through the ROS–HIF-1α axis triggered by ENO1–ACTB interaction-induced metabolic reprogramming. This unique pathway contributes significantly to the inflammation associated with *M. bovis* chronic infection.

ACTB, as a core component of the cytoskeleton, has seen its functional research expand from traditional roles in maintaining cell morphology and regulating motility to broader biological processes. On one hand, the expression level of ACTB directly impacts cell migration ability and morphological plasticity by modulating actin dynamics [40]; on the other hand, its interactions with other cytoskeletal proteins, such as cofilin and the Arp2/3 complex, form a complex regulatory network [41]. Notably, our findings reveal a novel functional duality of ACTB in the metabolic–immunological network; overexpression promotes an immunosuppressive microenvironment, whereas functional loss impairs pathogen clearance. This bidirectional regulatory capacity suggests that ACTB may act as a molecular switch, potentially mediating the transmission of metabolic stress signals through conformational changes. This concept aligns with known ACTB conformational dynamics [42] and warrants further structural investigation. This identifies ACTB as a key regulator mediating pathogen-induced metabolic reprogramming and subsequent inflammatory responses, a function distinct from its canonical roles in cell morphology and motility.

Compared to other pathogens, *M. bovis* exhibits unique metabolic manipulation strategies. Unlike *mycobacterium tuberculosis*, which suppresses glycolysis to evade immunity [43], and *Brucella*, which activates glycolysis to promote intracellular survival [44], the metabolic reprogramming induced by the ENO1–ACTB interaction in *M. bovis* possesses dual characteristics of energy hijacking and immune modulation. This synergistic model of “metabolic hijacking–inflammation manipulation” may be closely related to its chronic infection characteristics, providing a new mechanism for explaining the persistent infections caused by *M. bovis*. This study has systematically examined the molecular mechanisms by which the ENO1–ACTB axis regulates metabolic reprogramming. Based on these findings, future studies could focus on three key directions: first, the development of small molecule inhibitors targeting the ENO1–ACTB protein interaction interface will offer new avenues for targeted therapy; second, by integrating metabolic regulatory mechanisms with existing therapies, a systematic exploration of feasible multi-target combination treatment plans can be conducted; finally, establishing tissue-specific ACTB gene conditional knockout animal models will provide an important in vivo research platform for validating the function of this target in physiological and pathological processes.

## 5. Conclusions

This study systematically reveals the molecular mechanism by which *M. bovis* effector protein ENO1 regulates the metabolic–inflammatory network by targeting host cell ACTB. Experiments confirm that ENO1 induces glycolytic reprogramming (upregulation of Glut1, lactate accumulation, and ATP suppression) and ROS accumulation by upregulating ACTB expression, thereby activating the HIF-1α/IL-1β signaling axis and the pro-inflammatory factor network (IL-6/TNF-α). Structural analysis identifies the 117Glu and 372Arg of ACTB as key interaction sites, whose mutations significantly affect the binding capacity of ENO1. Functional studies indicate that the ENO1–ACTB axis establishes a ‘glycolysis–oxidative stress–inflammation’ cascade response, creating a stress microenvironment that inhibits the intracellular survival of *M. bovis*. These findings provide new insights into the understanding of pathogen–host metabolic interactions mediated by the ENO1–ACTB axis.

## Figures and Tables

**Figure 1 biomolecules-15-01107-f001:**
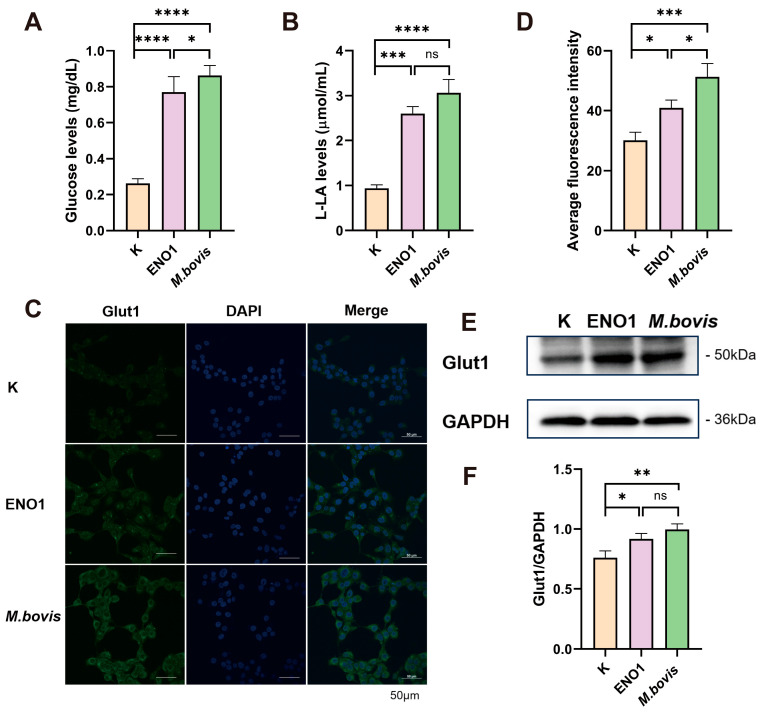
ENO1 overexpression and *M. bovis* infection enhance glycolysis in EBL cells. (**A**) Increased intracellular glucose concentration. EBL cells transfected with pcDNA3.1–ENO1 for 48 h (ENO1) or infected with *M. bovis* (MOI = 1000, 12 h) showed significantly elevated intracellular glucose levels compared to untreated controls (K). (**B**) Elevated lactate production. Lactate output was significantly higher in ENO1- and *M. bovis*-infected groups under identical conditions. (**C**) Enhanced membrane localization of Glut1. Confocal microscopy images (Green: Glut1; Blue: DAPI) demonstrate increased Glut1 enrichment on the plasma membrane of EBL cells following ENO1 or *M. bovis* infection. (**D**) Quantification of Glut1 membrane fluorescence intensity. Glut1 membrane signal intensity was significantly increased in ENO1 and *M. bovis* groups versus controls. (**E**) Upregulated total Glut1 protein expression. Western blot analysis confirmed increased Glut1 levels in ENO1 and *M. bovis*-infected cells (GAPDH loading control). Original Western blot images can be found in Appendix A. (**F**) Statistical analysis of Glut1 expression. Densitometric quantification of (**E**) normalized to control (K = 1). * All data presented herein represent the results from three separate experiments and are mean ± SD. Ns—not significant (*p* > 0.05), 0.01 < * *p* <0.05, ** *p* < 0.01, *** *p* < 0.001, and **** *p* < 0.0001.

**Figure 2 biomolecules-15-01107-f002:**
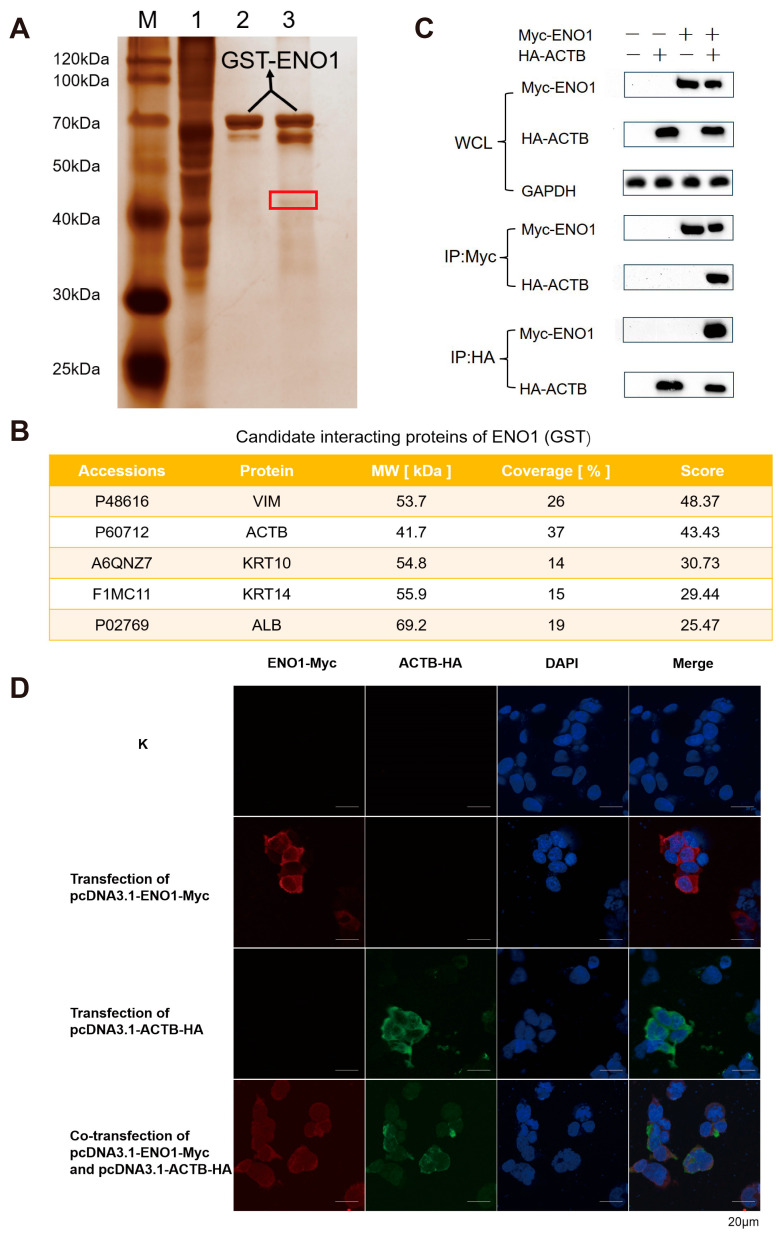
Validation of ENO1 interaction with host β-actin (ACTB). (**A**) GST pull-down screening of interacting proteins. Silver-stained SDS-PAGE shows specific bands (red box) in the experimental lane (Lane 3: ENO1-GST + EBL lysate) absent in controls (Lane 1: EBL lysate; Lane 2: ENO1-GST). M: Protein marker. (**B**) Mass spectrometry identification of candidate interactors. NanoLC-MS/MS analysis of the excised band (**A**, red box) identified top candidate proteins VIM, ACTB, KRT10, KRT14, ALB. (**C**) Co-immunoprecipitation (Co-IP) confirmation. Co-transfection of Myc-ENO1 and HA-ACTB in HEK293T cells. Anti-Myc antibody precipitated HA-ACTB (top), while anti-HA antibody precipitated Myc-ENO1 (bottom). Input controls shown. Original Western blot images can be found in Appendix A. (**D**) Subcellular co-localization of ENO1 and ACTB. HEK293T cells co-transfected with Myc-ENO1 (Red, Alexa Fluor 594) and HA-ACTB (Green, Alexa Fluor 488). Nuclei stained with DAPI (Blue). Yellow regions indicate cytoplasmic co-localization. Note: Panels (**A**,**B**) used EBL lysates; Panels (**C**,**D**) used HEK293T cells.

**Figure 3 biomolecules-15-01107-f003:**
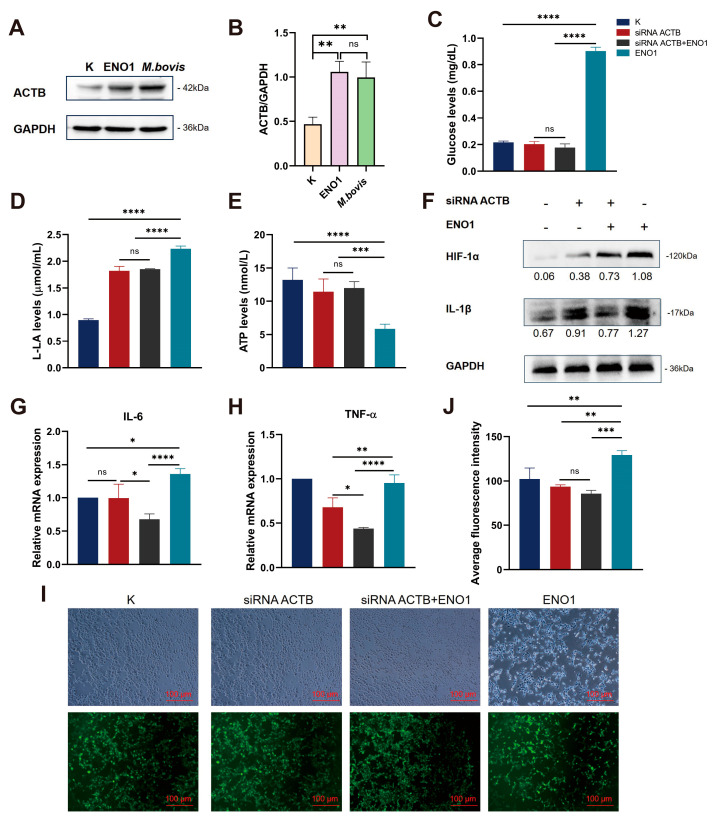
ENO1 mediates glycolytic reprogramming and inflammatory responses in EBL cells via ACTB upregulation. (**A**) Upregulated ACTB protein expression. Western blot shows significantly increased ACTB levels in ENO1 and *M. bovis*-infected EBL cells versus controls (K) (GAPDH loading control). Original Western blot images can be found in Appendix A. (**B**) Quantification of ACTB expression. Densitometric analysis of (**A**) normalized to control (K = 1). (**C**–**E**) ACTB knockdown reverses ENO1-induced metabolic alterations. Groups: ① Untreated (K); ② siRNA ACTB; ③ siRNA ACTB + ENO1; ④ ENO1. (**C**) Glucose concentration: Increased in ENO1 group, decreased in siRNA ACTB group, unchanged in siRNA ACTB +ENO1 group. (**D**) Lactate production: Elevated in ENO1 group, reduced in siRNA ACTB group, unchanged in siRNA ACTB + ENO1 group. (**E**) ATP levels: Suppressed in ENO1 group, restored in siRNA ACTB group, unchanged in siRNA ACTB + ENO1 group. (**F**) Upregulated HIF-1α and IL-1β protein expression. ENO1 increased both proteins; siRNA ACTB blocked this effect (GAPDH-normalized values). Original Western blot images can be found in Appendix A. (**G**,**H**) Increased IL-6 and TNF-α mRNA expression. qRT-PCR shows elevated IL-6 (**G**) and TNF-α (**H**) transcription in ENO1 group; siRNA ACTB attenuated this response (2^−ΔΔCt^ method, K = 1). (**I**,**J**) Enhanced ROS accumulation. DCFH-DA fluorescence images (**I**) and quantification (**J**) demonstrate increased ROS in ENO1 group, attenuated by siRNA ACTB. * All data presented herein represent the results from three separate experiments and are mean ± SD. Ns—not significant (*p* > 0.05), 0.01 < * *p* < 0.05, ** *p* < 0.01, *** *p* < 0.001, and **** *p* < 0.0001.

**Figure 4 biomolecules-15-01107-f004:**
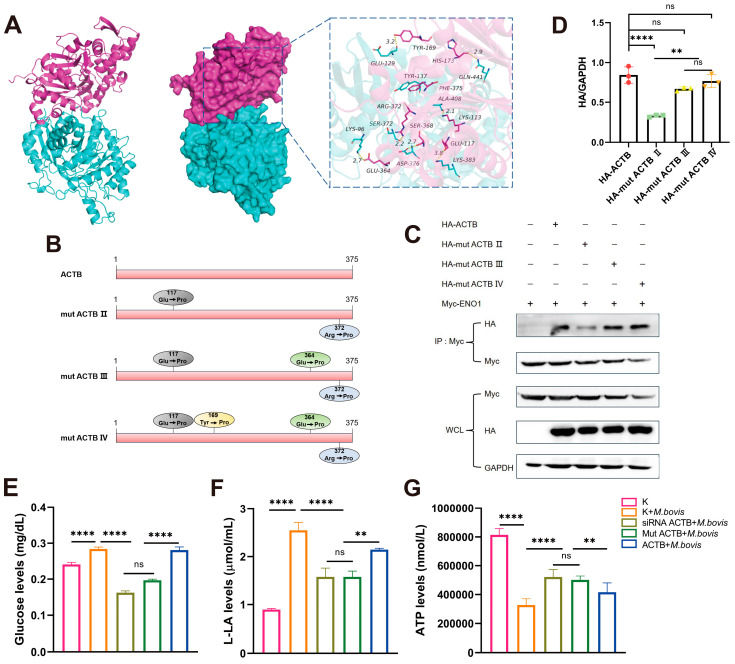
ENO1–ACTB interaction and glycolytic regulation depend on ACTB Glu117 and Arg372 residues. (**A**) Molecular docking model of ENO1–ACTB complex. ENO1 (cyan) interacts with ACTB (red) via hydrogen bonds (e.g., ACTB Arg372) and salt bridges (e.g., ACTB Glu117). (**B**) ACTB point mutant design. Constructs: double mutant (E117P/R372P), triple mutant (E117P/E364P/R372P), and quadruple mutant (E117P/Y169P/E364P/R372P). (**C**) Co-IP validation of interaction strength. HEK293T cells co-transfected with Myc-ENO1 and WT/mutant HA-ACTB. Anti-Myc precipitation revealed significantly weakened binding with double mutant (mut ACTB III). Original Western blot images can be found in Appendix A. (**D**) Quantification of interaction intensity. WB band analysis normalized to WCL GAPDH shows reduced binding efficiency of double mutant. (**E**–**G**) ACTB mutants reverse *M. bovis*-induced metabolic changes. Groups: ① Untreated (K); ② *M. bovis* infection (*M. bovis)*; ③ siRNA ACTB + *M. bovis*; ④ mut ACTB + *M. bovis*; ⑤ ACTB + *M. bovis*. (**E**) Glucose concentration: increased in *M. bovis* group; partially reversed in siRNA ACTB /mut ACTB + *M. bovis* groups. (**F**) Lactate production: elevated in *M. bovis* group; partially suppressed in siRNA ACTB/mut ACTB + *M. bovis* groups. (**G**) ATP levels: reduced in *M. bovis* group; partially restored in siRNA ACTB /mut ACTB + *M. bovis* groups. * All data presented herein represent the results from three separate experiments and are mean ± SD. Ns—not significant (*p* > 0.05), ** *p* < 0.01, and **** *p* < 0.0001.

**Figure 5 biomolecules-15-01107-f005:**
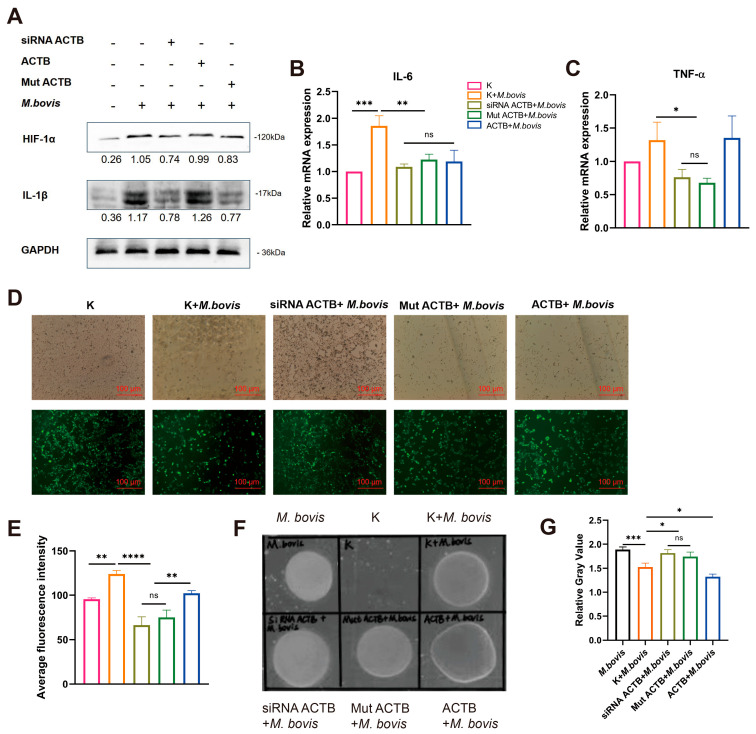
ACTB-mediated inflammatory responses regulate M. bovis intracellular survival. Groups (EBL cells): ① Untreated (K); ② *M. bovis* infection (*M. bovis)*; ③ siRNA ACTB + *M. bovis*; ④ mut ACTB + *M. bovis*; ⑤ ACTB + *M. bovis*. (**A**) HIF-1α and IL-1β protein expression. Significantly upregulated in M. bovis group. Partially suppressed in siRNA ACTB/mut ACTB + *M. bovis* groups (GAPDH-normalized values). Original Western blot images can be found in Appendix A. (**B**,**C**) IL-6 and TNF-α mRNA expression. qRT-PCR shows elevated IL-6 (**B**) and TNF-α (**C**) transcription in *M. bovis* group, attenuated in siRNA ACTB/mut ACTB + *M. bovis* groups (2^−ΔΔCt^ method, K = 1). (**D**,**E**) ROS accumulation. DCFH-DA images (**D**) and fluorescence quantification (**E**) show increased ROS in *M. bovis* group, attenuated in siRNA ACTB/mut ACTB + *M. bovis* groups. (**F**) Representative bacterial colonies. *M. bovis*-infected cell lysates plated on PPLO agar (7-day incubation). (**G**) Pathogen burden analysis. Colony gray value quantification revealed significantly higher bacterial loads in siRNA ACTB/mut ACTB + *M. bovis* groups versus *M. bovis* group, and lowest loads in ACTB + *M. bovis* group. * All data presented herein represent the results from three separate experiments and are mean ± SD. Ns—not significant (*p* > 0.05), 0.01 < * *p* < 0.05, ** *p* < 0.01, *** *p* < 0.001, and **** *p* < 0.0001.

**Figure 6 biomolecules-15-01107-f006:**
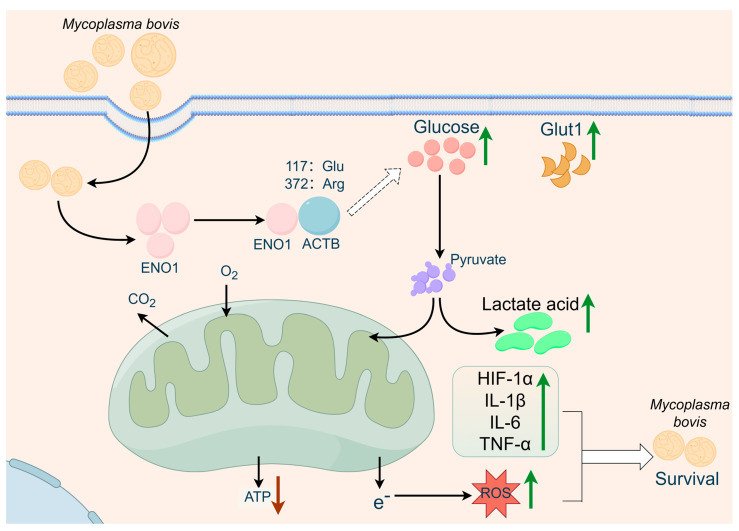
Proposed model of ENO1–ACTB interaction-mediated metabolism-inflammation interplay in anti-infection defense based on the multi-group experimental results of this study. We propose that the mycoplasma bovis effector protein ENO1 targets host ACTB to regulate a dual regulatory axis involving glycolytic reprogramming and inflammatory response, thereby restricting intracellular persistent infection (figure created with FigDraw 2.0).

## Data Availability

The original contributions presented in this study are included in the article/Appendix A. Further inquiries can be directed to the corresponding authors.

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
