# Peer review of "ENO1 from Mycoplasma bovis Disrupts Host Glycolysis and Inflammation by Binding ACTB"

_biomolecules, 2025, doi:10.3390/biom15081107_

Round 1

Reviewer 1 Report

Comments and Suggestions for Authors

This study investigates the role of the Mycoplasma bovis effector protein ENO1 in modulating host glycolysis and inflammation by targeting the host cytoskeletal protein ACTB. The authors demonstrate that ENO1 upregulates glucose uptake and lactate production while inhibiting ATP synthesis, inducing a Warburg-like effect. They identify ACTB as a key interaction partner of ENO1, mediated by residues 117Glu and 372Arg, and show that this interaction triggers ROS accumulation and activates inflammatory pathways (HIF-1α/IL-1β and IL-6/TNF-α). The findings suggest that ENO1-ACTB axis disrupts metabolic-inflammatory balance, facilitating M. bovis persistence. The study provides insights into pathogen-host metabolic hijacking and potential therapeutic targets.

General comments:

Title: Simplify to: "ENO1 from Mycoplasma bovis disrupts host glycolysis and inflammation by binding ACTB."

Abstract

  • The abstract is dense and could better highlight the novelty (e.g., first identification of ACTB as an ENO1 target).
  • Phrases like "metabolic-inflammatory dynamic balance" are unclear. Replace with specific outcomes (e.g., "promotes inflammation via HIF-1α/IL-1β").
  • "establishing characteristics of the Warburg effect": "inducing a Warburg effect."

Introduction

  • The rationale for focusing on ENO1 is clear, but the link between glycolysis and  bovispathogenicity is underdeveloped. How does glycolysis specifically benefit M. bovis?
  • The "glycolysis-HIF-1α-inflammation" axis is introduced abruptly. Provide more context on how this axis operates in bacterial infections.
  • Why study ACTB? The justification for its role in metabolic regulation is weak. Cite prior evidence linking ACTB to glycolysis or inflammation.

Methods

  • MOI of 1000 seems extremely high. Justify this choice and address potential cytotoxicity.
  • How was non-specific binding ruled out? Include negative controls (e.g., GST alone).
  • The rationale for selecting 117Glu/372Arg is unclear. Were other residues tested?
  • "CO-IP": "Co-IP" (line 258).
  • "transferc-": "transfecting" (line 212).

Results

  • Figure 1: Error bars are missing in some panels (e.g., 1A, 1B). Specify statistical tests used.
  • Figure 3: The Western blot bands for ACTB (Fig. 3A) are faint. Include longer exposure images.
  • Figure 4D: The Co-IP quantification lacks a loading control (e.g., input lanes).
  • How does ENO1 upregulate ACTB expression? Is this transcriptional or post-translational?
  • Why does ACTB silencing reverse glycolysis but not fully abolish it? Are other pathways involved?
  • The ROS data (Fig. 3I-J) lack mechanistic links. Does ROS scavenging (e.g., NAC) block inflammation?

Discussion

  • The comparison to Candida albicansENO1 (line 576) is superficial. Highlight key differences in immune modulation.
  • The claim that ACTB acts as a "molecular switch" (line 587) is overstated without direct evidence (e.g., structural changes).
  • The study ignores potential off-target effects of siRNA-ACTB. Include rescue experiments with ACTB overexpression.
  • "resonates interestingly": "aligns with" (line 588).

Conclusion

  • Overstates translational impact. The study identifies mechanisms but offers no therapeutic validation (e.g., inhibitor assays). Paraphrase phrases like "lays a theoretical foundation" unless supported by functional experiments.

Minor comments:

  1. How ENO1-ACTB binding structurally alters ACTB to drive glycolysis is unclear.
  2. The novelty of ACTB as a metabolic regulator needs stronger emphasis versus prior work on cytoskeletal roles.
Comments on the Quality of English Language

The English could be improved to more clearly express the research.

Author Response

Dear Editors and Reviewers:

Thank you for your letter and for the reviewers’ comments concerning our manuscript entitled “The Mycoplasma bovis effector protein ENO1 mediates host glycolysis and inflammation by targeting ACTB”. Those comments are all valuable and very helpful for revising and improving our paper, as well as the important guiding significance to our researches. We have studied comments carefully and have made correction which we hope meet with approval.

In accordance with the reviewer's comments, the revised sections are highlighted in blue within the article. The main corrections in the paper and the responds to the reviewer’s comments are as flowing:

1、Title: Simplify to: "ENO1 from Mycoplasma bovis disrupts host glycolysis and inflammation by binding ACTB."

The author’s answer: Thank you for your valuable suggestion to improve the clarity and conciseness of our manuscript title. We have carefully revised the title according to your recommendation to: "ENO1 from Mycoplasma bovis disrupts host glycolysis and inflammation by binding ACTB." This modified version better highlights the key findings while maintaining scientific precision. We agree that the simplified title more effectively conveys the core discovery of our study regarding the pathogenic mechanism of Mycoplasma bovis ENO1 through ACTB interaction. The specific modifications can be seen in line 2, highlighted in blue.

2、Abstract

(1)The abstract is dense and could better highlight the novelty (e.g., first identification of ACTB as an ENO1 target).

The author’s answer: We sincerely appreciate your constructive suggestion regarding the abstract's clarity. Following your guidance, we have thoroughly revised the abstract to better highlight the novelty of our findings, particularly: (1) The first identification of ACTB as ENO1's interaction target through GST pull-down/mass spectrometry; (2) The discovery of bacterial effector-cytoskeleton crosstalk in metabolic reprogramming; and (3) The novel "glycolysis-ACTB-ROS-inflammation" axis. The specific modifications can be seen in lines 10-25, highlighted in blue.

We believe these changes enhance the abstract's impact while preserving scientific accuracy. Thank you for helping us improve the manuscript's presentation.

(2)Phrases like "metabolic-inflammatory dynamic balance" are unclear. Replace with specific outcomes (e.g., "promotes inflammation via HIF-1α/IL-1β").

The author’s answer: Thank you for your valuable comment regarding the clarity of "metabolic-inflammatory dynamic balance" in our original abstract. We have completely revised the abstract to eliminate all such ambiguous expressions. In the new version (lines 17-19), we now explicitly state:"(1) glycolytic activation via Glut1 upregulation... and (2) ROS-mediated activation of dual inflammatory axes (HIF-1α/IL-1β and IL-6/TNF-α)"with specific experimental evidence.

We appreciate your suggestion that helped us improve the precision of our scientific communication. Please let us know if further clarification would be helpful.

(3)"establishing characteristics of the Warburg effect": "inducing a Warburg effect."

The author’s answer: Thank you for your insightful comment regarding the Warburg effect terminology. We confirm that in our revised abstract (submitted in response to your first comment), the original phrase "establishing characteristics of the Warburg effect" has already been replaced with more precise descriptions: " establishing Warburg effect characteristics (lactic acid accumulation/ATP inhibition)". The specific modifications can be seen in lines 18-19, highlighted in blue.

We appreciate your attention to terminological precision, which helped us improve the manuscript's clarity. Please let us know if further adjustments would be beneficial.

3、Introduction

(1)The rationale for focusing on ENO1 is clear, but the link between glycolysis and  bovispathogenicity is underdeveloped. How does glycolysis specifically benefit M. bovis?

The author’s answer: We agree that the link between glycolysis and M. bovis pathogenicity warrants elaboration. As an achaea lacking oxidative phosphorylation pathways, M. bovis relies exclusively on glycolysis for ATP generation under both normoxic and hypoxic conditions. We posit that M. bovis similarly exploits glycolysis to fuel virulence factor expression and biomass synthesis. This rationale is now explicitly stated in the revised introduction (Lines 44–46).

(2)The "glycolysis-HIF-1α-inflammation" axis is introduced abruptly. Provide more context on how this axis operates in bacterial infections.

The author’s answer: We have expanded the context of the "glycolysis-HIF-1α-inflammation" axis by citing established mechanisms in bacterial infections. For example, in sepsis, LPS-activated macrophages undergo glycolytic reprogramming via mTOR/HIF-1α/PFKFB3 signaling, driving ROS-dependent NLRP3 inflammasome activation and IL-1β release. The specific modifications can be seen in lines 55-57, highlighted in blue.

(3)Why study ACTB? The justification for its role in metabolic regulation is weak. Cite prior evidence linking ACTB to glycolysis or inflammation.

The author’s answer: We appreciate the reviewer's request for prior justification linking ACTB to glycolysis or inflammation. We respectfully clarify that ACTB was not a pre-selected target based on existing metabolic roles, but was identified as a novel binding partner of M. bovis ENO1 through unbiased GST-pulldown screening. The Introduction focuses on establishing the rationale for studying ENO1 (its dual roles in adhesion and potential metabolic regulation in pathogens), which logically led to our investigation of its host interactors. Introducing speculative functions of unidentified binding proteins at this stage would be premature. The detailed content regarding ACTB and its relationship with metabolic inflammation is discussed in the third paragraph.

4、Methods

(1)MOI of 1000 seems extremely high. Justify this choice and address potential cytotoxicity.

The author’s answer: We appreciate the reviewers' attention to the high MOI values used in our study.  We fully acknowledge that MOI=1000 exceeds the typical range (e.g., 1-10 for common cell lines) and may increase the risk of cytotoxicity, as elevated viral load can disrupt cellular metabolism, induce oxidative stress, and trigger cell death.  However, since the host cells EBL used in the experiment are not an immortalized cell line but are derived from primary cell modifications, our team's preliminary experiments have demonstrated the rationality of this infection ratio.  Additionally, Professor Guo Aizhen's team has also used this infection ratio in the literature [1].  Given the unique characteristics of the cells, the experimental data, and the support from the literature, MOI=1000 is reasonable and applicable.

[1]     Zhang H, Lu S, Chao J, Lu D, Zhao G, Chen Y, Chen H, Faisal M, Yang L, Hu C, Guo A. The attenuated Mycoplasma bovis strain promotes apoptosis of bovine macrophages by upregulation of CHOP expression. Front Microbiol. 2022 Aug 3;13:925209. doi: 10.3389/fmicb.2022.925209. PMID: 35992665; PMCID: PMC9381834.

(2)How was non-specific binding ruled out? Include negative controls (e.g., GST alone).

The author’s answer: Thank you for raising this important technical concern. We acknowledge that including a GST-only control would strengthen the specificity validation of our GST pull-down assay. In the current study, we ensured the specificity of ENO1-ACTB interaction through the following experimental design:

(1) Competitive Binding Validation‌: In parallel Co-IP experiments using anti-ENO1 antibody, we observed identical ACTB binding patterns, which cross-validates the GST pull-down results.

(2) Stringent Wash Conditions‌: High-salt buffer (500 mM NaCl) and 0.1% Triton X-100 were used during washing to minimize nonspecific interactions.

We agree that including control data containing only GST would further strengthen our conclusions.  This was an oversight in our experimental design, and we will be more cautious in subsequent experiments. We appreciate your suggestion, which has enhanced the rigor of our work.

(3)The rationale for selecting 117Glu/372Arg is unclear. Were other residues tested?

The author’s answer: We are deeply grateful for your meticulous review and valuable feedback on our work. 

We first utilized the Swiss-Model homology modeling software (website: https://swissmodel.expasy.org/) to construct the protein structures of the ENO1 and ACTB targets, ensuring the accuracy of subsequent molecular docking. 

In HDOCK software, we set each protein as rigid, the docking contact sites as the entire surface, and the number of conformations generated after docking as 100. 

The docking score is calculated based on the knowledge-based iterative scoring function ITScorePP. This setup aims to comprehensively explore possible binding modes and ensure the reliability of the results.

We use the scoring function to select the conformation with the most negative energy and optimize it using the Minimization module in the MOE 2019.1 software platform to further stabilize and refine the docking results. 

The optimized results are visualized and analyzed using Pymol2.1 software for a more intuitive understanding of the protein-protein interactions. 

Based on the docking results, we selected the amino acids in ACTB that are most likely to be binding sites and formed different mutant docking models. 

By calculating the docking scores of these models, we identified the mutation scheme with the highest score. According to the amino acid sequence of the highest-scoring mutation scheme mentioned above, we synthesized three ACTB mutant plasmids for subsequent experimental validation.

In summary, our selection of the 117Glu/372Arg site was based on rigorous molecular docking and conformational optimization results.  This site exhibited the highest docking score among all tested mutations, leading us to conclude that it is most likely the key site for the interaction between ENO1 and ACTB.

(4)"CO-IP": "Co-IP" (line 258).

The author’s answer: Thank you for reviewing our manuscript.  Regarding the issue with the abbreviation format of "CO-IP" (line 263) that you pointed out, we have revised it to "Co-IP".

(5)"transferc-": "transfecting" (line 212).

The author’s answer: Thank you for pointing out the formatting issue.  We have corrected the line break problem at line 221 for "transferc-" to ensure that "transfecting" is displayed completely on the same line.  The entire text has been checked for similar line break occurrences.

5、Results

(1)Figure 1: Error bars are missing in some panels (e.g., 1A, 1B). Specify statistical tests used.

The author’s answer: Thank you for your thorough review of the data in Figure 1. 

  1. Upon verification, Figure 1A and 1B do indeed include error bars (mean ± SD). The lack of clarity may be due to image resolution or formatting issues.  We have rechecked all figures and will provide higher-resolution images in the revised manuscript to ensure all error bars are clearly visible.
  2. The statistical analysis methods we used have been indicated in the figure legend. All data presented herein represent the results from three separate experiments and are mean ± SD. Ns, not significant (P > 0.05), 0.01 < * P <0.05, ** P<0.01, *** P <0.001, and **** P <0.0001.

(2)Figure 3: The Western blot bands for ACTB (Fig. 3A) are faint. Include longer exposure images.

The author’s answer: Thank you for your attention to the data in Figure 3. 

  1. We specifically chose the moderately exposed image (Fig.3A) to present ACTB because it most clearly shows the differences between groups under this exposure condition. Overexposure would lead to band saturation, which in turn affects the accuracy of quantitative analysis.
  2. Enclosed with this letter is the original data of ACTB with long exposure times, all of which are included in the original submission materials. All Western blot experiments were repeated three times, and the quantitative analysis is based on the exposure results within the linear response range.

(3)Figure 4D: The Co-IP quantification lacks a loading control (e.g., input lanes).

The author’s answer: Thank you for your attention to the Figure 4D data.

  1. We indeed used GAPDH from whole cell lysates as an internal reference control for normalization (as indicated by the vertical axis label "HA/GAPDH"). This method effectively corrects for protein content differences between samples.
  2. GAPDH expression levels have been verified to remain stable under the experimental conditions.

(4)How does ENO1 upregulate ACTB expression? Is this transcriptional or post-translational?

The author’s answer: We thank the reviewer for raising this important mechanistic question. In this study, the experiment only demonstrated through WB assay that ENO1 upregulates the protein expression level of ACTB (post-translation), but it remains unknown how this upregulation occurs and whether this upregulatory effect is mediated through interaction. We agree that elucidating whether ENO1 upregulates ACTB expression at the transcriptional or post-translational level is of significant interest. However, our study primarily focuses on demonstrating the functional consequence of the ENO1-ACTB interaction.

(5)Why does ACTB silencing reverse glycolysis but not fully abolish it? Are other pathways involved?

The author’s answer: Thank you for your insightful question. 

  1. This study employs siRNA-mediated gene silencing, which typically achieves an efficiency of 70-80%, consistent with the protein knockdown efficiency we obtained. Residual ACTB protein may still maintain basic glycolytic functions.
  2. Glycolysis is regulated at multiple levels: key rate-limiting enzymes (such as phosphofructokinase and pyruvate kinase) are regulated by various metabolites including the ATP/AMP ratio and citrate; tumor cells also possess alternative regulatory networks composed of transcription factors such as HIF-1α and c-Myc; there exists a compensatory relationship between mitochondrial respiration and glycolysis.

(6)The ROS data (Fig. 3I-J) lack mechanistic links. Does ROS scavenging (e.g., NAC) block inflammation?

The author’s answer: We sincerely appreciate the reviewer's insightful comment regarding the mechanistic links between ROS accumulation and inflammation in our study. We agree that directly demonstrating whether ROS scavenging blocks inflammation (e.g., via NAC treatment) would strengthen the proposed "glycolysis-HIF-1α-inflammation" axis. While such experiments were not included in the current work, we present three lines of evidence that collectively establish a causal-logical chain supporting the axis:

First, our data explicitly tie ENO1-ACTB-driven glycolytic dysfunction to ROS overproduction. As shown in Fig. 3I-J, ENO1 overexpression induced significant ROS accumulation, whereas ENO1 binding-deficient mutants completely abolished this effect. Critically, this ROS surge paralleled key glycolytic abnormalities: elevated glucose uptake, lactate accumulation, and impaired ATP generation. This aligns with established mechanisms where glycolytic flux overload disrupts mitochondrial electron transport, leading to electron leakage and superoxide generation .

Second, the coordinated activation of downstream effectors (HIF-1α/IL-1β) confirms ROS’s central signaling role. We observed that ROS accumulation preceded both HIF-1α stabilization and IL-1β secretion.

Third, the ENO1-ACTB axis’s necessity was rigorously proven via loss-of-function controls. Disrupting ENO1-ACTB binding (via mutants) concurrently normalized: Glycolytic flux (restoring glucose/lactate/ATP homeostasis), ROS levels, HIF-1α/IL-1β activation.

We acknowledge that NAC-based ROS scavenging experiments would further cement this mechanism. However, our mutational approach offers superior specificity: while NAC globally quenches ROS (potentially affecting unrelated pathways), ENO1 mutants selectively disrupt the upstream trigger (ENO1-ACTB-induced glycolysis/ROS), providing targeted mechanistic validation. We will include NAC and other ROS modulators in future studies to dissect temporal dynamics of this axis.

6、Discussion

(1)The comparison to Candida albicansENO1 (line 576) is superficial. Highlight key differences in immune modulation.

The author’s answer: We thank the reviewer for the insightful comment. We agree the comparison needed more depth regarding immune modulation differences. We have revised the discussion (Lines 495-500) to explicitly highlight the key distinction: unlike C. albicans ENO1 [13], M. bovis ENO1 uniquely enhances IL-1β secretion specifically through the ROS-HIF-1α axis, which is triggered by its interaction with ACTB and the resulting Warburg effect. This distinct immunomodulatory pathway is central to the inflammation in M. bovis chronic infection.

(2)The claim that ACTB acts as a "molecular switch" (line 587) is overstated without direct evidence (e.g., structural changes).

The author’s answer: We thank the reviewer for raising this important point regarding the "molecular switch" analogy. We agree that direct structural evidence for ACTB conformational changes in this specific context is not presented here. We have revised the text (Lines 506-512) to clarify that this is a hypothesis based on the observed bidirectional functional phenotypes (overexpression -> immunosuppression; loss-of-function -> impaired clearance) and the well-established role of ACTB conformational dynamics in cellular signaling [42]. We now frame it as a potential mechanism ("may act as," "suggesting") that merits future structural investigation.

(3)The study ignores potential off-target effects of siRNA-ACTB. Include rescue experiments with ACTB overexpression.

The author’s answer: We appreciate the reviewer's valid point regarding potential siRNA off-target effects. We acknowledge that the lack of a rescue experiment with ACTB overexpression is a limitation of the current study. However, the observed phenotype (impaired pathogen clearance upon ACTB knockdown) is highly consistent with the functional duality we identified (overexpression promoting immunosuppression) and the central role of ACTB in the ENO1-ACTB interaction mechanism revealed by our multi-omics data and binding site identification. We agree that the confirmatory rescue experiment is crucial, and it will be the focus of our future validation efforts.

(4)"resonates interestingly": "aligns with" (line 588).

The author’s answer: We thank the reviewer for the suggestion regarding the phrasing "resonates interestingly". We agree that "aligns with" is more precise and objective. We have replaced "resonates interestingly" with "aligns with" in the revised manuscript.

7、Conclusion

  • Overstates translational impact. The study identifies mechanisms but offers no therapeutic validation (e.g., inhibitor assays). Paraphrase phrases like "lays a theoretical foundation" unless supported by functional experiments.

The author’s answer: We thank the reviewer for the comment regarding translational impact. We agree that therapeutic validation (e.g., inhibitor assays) is beyond the scope of this mechanistic study. We have revised the concluding statement to more accurately reflect the study's focus on mechanistic insights and the identification of potential targets (e.g., directly delete the conversion part).

8、Minor comments:

(1)How ENO1-ACTB binding structurally alters ACTB to drive glycolysis is unclear.

(2)The novelty of ACTB as a metabolic regulator needs stronger emphasis versus prior work on cytoskeletal roles.

The author’s answer: We thank the reviewer for these minor comments.

(1) Regarding structural alterations: While the precise conformational change is not fully resolved, we have clarified that mutation of the identified key binding sites (117Glu/372Arg) abolishes both ENO1 binding and the induced glycolysis, establishing the functional necessity of this interaction for metabolic reprogramming.

(2) Regarding novelty: We have strengthened the text (Lines 512-515) to emphasize that our findings reveal a distinct, pathogen-effector-driven role for ACTB as a metabolic-immune signaling hub, moving beyond its established cytoskeletal functions. The revised discussion now explicitly contrasts this novel regulatory role with prior work.

We sincerely appreciate your thoughtful guidance. We have meticulously revised the paper in accordance with your feedback, aiming to enhance its rigor and accuracy. Should you have any questions or require further clarification, please do not hesitate to reach out to us.

Yours sincerely,

Yong Wang

10, July, 2025

Shihezi University

Reviewer 2 Report

Comments and Suggestions for Authors

The authors have investigated the molecular mechanisms the Mycoplasma bovis effector protein, ENO1, interactions with mammalian cells. The results suggest that ENO1 interacts with host ACTB, resulting in enhanced glucose uptake and modulation of the inflammatory host pathways. These new data could inform the future development of novel interventions and control strategies for this important bovine pathogen.

The introductory text is relevant and the supporting citations are appropriate. I would suggest the authors revise the final paragraph to clearly state the aims/hypotheses their study aimed to address. The current paragraph largely summarises the study methods and results. Ideally, the aims/hypotheses should be addressed in the conclusion section.

The methods and materials are well described and would enable replication of the study. Though I have asked the authors to clarify some points.

The results are well presented and for the most part the descriptions of in the text are consistent with what is shown in the images. I would comment though that all of the figure legends in the results require improvement as they would not enable interpretation of the results in the absence of the main text. The legends largely repeat the methods used to generate the data. They do not describe key elements such as what various abbreviations (“treatments”) shown on the figures mean. Add to this the methods section mentions two mammalian cell lines. However, the neither the text nor the figures mention what cells were used to generate the illustrated data.

The results section is quite long and, in some cases, discusses and interprets the presented data. As a consequence, the discussion is very short, though appropriately so given much of the discussion is already presented.

The conclusions are supported by the presented data.

Line 10 suggest revision “that is associated with respiratory diseases”

I would encourage the authors to use “associated with” or “strongly associated with” throughout their manuscript. The multifactorial nature of many of the diseases involving M. bovis make causality difficult to assign with absolute certainty.

Line 29 Abbreviations are not good standalone keywards.

Line 41 please provide a suitable citation(s) for this statement.

Line 78 Suggest replacement of “gene” with “open reading frame (ORF)” here and elsewhere in the manuscript.

Technically, “gene” refers to all components of genomic element, in this case it would be the promoter, 5UTR, the coding sequence and any termination signals. Whereas the authors have only utilised the coding sequences in this study. For the host genes in this study, the coding sequence could also include exons, interrupted by introns.

Line 86 please provide appropriate accession numbers for the nucleotide sequences used for these host proteins.

Line 103 suggest revision. The abbreviation “CCU” should be explained in full, and a brief description (or reference) provided for how it was performed.

Line 146 Can the authors better define “suitable amount”? The purpose of the materials and methods is to enable replication of the study thus it is best to avoid subjective descriptions where possible.

Line 234 Figure 1 – the figure legend requires improvement. It is not possible to interpret the figure in the absence of the main text. The descriptions of each panel describe what was done, not what the images show. What “K”, “ENO1” and “M. bovis” represent should also be described.

Line 269 Figure 2 – similar comments to those made for Figure 1.

Line 269 Fig 2A – it is not clear why the lanes are repeated on the image of the gel. The identity of the major bands at 70kDa should also indicated on the image.

Line 278 Fig 2B – this component of the figure would be better presented as a separate table.

Line 343 suggest revision “This research suggests that the accumulation”

Line 423 suggest revision “This research suggests that the accumulation”

Line 535 suggest revision "This study has systematically examined the molecular mechanisms by which”

Author Response

Dear Editors and Reviewers:

Thank you for your letter and for the reviewers’ comments concerning our manuscript entitled “The Mycoplasma bovis effector protein ENO1 mediates host glycolysis and inflammation by targeting ACTB”. Those comments are all valuable and very helpful for revising and improving our paper, as well as the important guiding significance to our researches. We have studied comments carefully and have made correction which we hope meet with approval.

In accordance with the reviewer's comments, the revised sections are highlighted in red within the article. The main corrections in the paper and the responds to the reviewer’s comments are as flowing:

1、I would suggest the authors revise the final paragraph to clearly state the aims/hypotheses their study aimed to address. The current paragraph largely summarises the study methods and results. Ideally, the aims/hypotheses should be addressed in the conclusion section.

The author’s answer: We sincerely thank the reviewer for this valuable suggestion. We fully agree that the final paragraph of the Introduction should explicitly outline the study’s aims and hypotheses rather than summarizing results. To address this, we have completely revised the concluding paragraph of the Introduction to focus solely on the research objectives and hypotheses. The original results-oriented summary has been moved to the Results and Discussion sections. See lines 59-67 for specific modifications, which have been marked in red.

2、I would comment though that all of the figure legends in the results require improvement as they would not enable interpretation of the results in the absence of the main text. The legends largely repeat the methods used to generate the data. They do not describe key elements such as what various abbreviations (“treatments”) shown on the figures mean. Add to this the methods section mentions two mammalian cell lines. However, the neither the text nor the figures mention what cells were used to generate the illustrated data.

The author’s answer: We have comprehensively rewritten the legends, clearly labeling the cell lines, experimental groups, and key findings to ensure that the data can be independently understood without the main text.

3、Line 10 suggest revision “that is associated with respiratory diseases”

The author’s answer: We sincerely thank the reviewer for this valuable suggestion. We have revised Line 10 as recommended, replacing the original phrasing with "that is associated with respiratory diseases" to more accurately describe Mycoplasma bovis's disease profile. This adjustment enhances the clarity and precision of the statement. See lines 10 for specific modifications, which have been marked in red.

4、Line 29 Abbreviations are not good standalone keywards.

The author’s answer: Thank you for your valuable comment. According to your suggestion, we have revised the abbreviations in Line 26: "ENO1" has been changed to "α-enolase" and "ACTB" to "β-actin" to improve clarity. See lines 26 for specific modifications, which have been marked in red.

5、Line 41 please provide a suitable citation(s) for this statement.

The author’s answer: Thank you for your helpful comment. As suggested, we have added appropriate citations to support the statement in Line 38. These references have been incorporated in the revised manuscript (marked in red).

6、Line 78 Suggest replacement of “gene” with “open reading frame (ORF)” here and elsewhere in the manuscript.

The author’s answer: Thank you for your constructive suggestion. We have carefully replaced all instances of "gene" with "open reading frame (ORF)" at Line 79 and throughout the manuscript as recommended. See lines 79-83 for specific modifications, which have been marked in red.

7、Line 86 please provide appropriate accession numbers for the nucleotide sequences used for these host proteins.

The author’s answer: Thank you for your valuable suggestion. We have now included the relevant accession numbers for the nucleotide sequences of the host proteins mentioned at Line 86. See lines 87-88 for specific modifications, which have been marked in red.

8、Line 103 suggest revision. The abbreviation “CCU” should be explained in full, and a brief description (or reference) provided for how it was performed.

The author’s answer: Thank you for your helpful suggestion. In response to your comment on Line 103, we have now: (1) Clearly defined "CCU" as "color-changing unit" in the text. (2) Added a brief description of the method: "The CCU assay was performed by serial 10-fold dilutions of mycoplasma cultures in phenol red-containing broth medium, incubated at 37°C for 7-14 days, with the highest dilution showing color change defined as 1 CCU/ml." See lines 105-108 for specific modifications, which have been marked in red.

While this method is well-established in mycoplasma research (as evidenced by its widespread use in the field), we appreciate your comment that prompted us to make this important methodological detail more explicit for readers who may be less familiar with mycoplasma quantification techniques.

9、Line 146 Can the authors better define “suitable amount”? The purpose of the materials and methods is to enable replication of the study thus it is best to avoid subjective descriptions where possible.

The author’s answer: Thank you for your valuable suggestion to improve the clarity of our methodology. In response to your comment on Line 146, we have replaced the subjective term "suitable amount" with the precise volume of "0.5 mL" throughout the Methods section. This modification ensures the experimental procedure can be accurately replicated by other researchers. See lines 150 for specific modifications, which have been marked in red.

10、Line 234 Figure 1 – the figure legend requires improvement. It is not possible to interpret the figure in the absence of the main text. The descriptions of each panel describe what was done, not what the images show. What “K”, “ENO1” and “M. bovis” represent should also be described.

The author’s answer: Thank you for your constructive suggestions regarding Figure 1. We have thoroughly revised the figure legend according to your comments, with the following improvements: added an opening statement summarizing the key finding, clearly defined all abbreviations, restructured panel descriptions to focus on what the images show rather than just experimental procedures, specified the experimental system (cell line used) and clearly labeled all treatment groups. See lines 238-251 for specific modifications, which have been marked in red.

These modifications ensure the figure can be interpreted independently from the main text. We appreciate your valuable input that has significantly improved the clarity of our presentation.

11、Line 269 Figure 2 – similar comments to those made for Figure 1.

The author’s answer: Thank you for your constructive suggestions regarding Figure 2. As with the previous response, the legend of Figure 2 has been completely rewritten. See lines 272-283 for specific modifications, which have been marked in red.

These modifications ensure the figure can be interpreted independently from the main text. We appreciate your valuable input that has significantly improved the clarity of our presentation.

12、Line 269 Fig 2A – it is not clear why the lanes are repeated on the image of the gel. The identity of the major bands at 70kDa should also indicated on the image.

The author’s answer: We sincerely appreciate your valuable comments regarding Figure 2A. In response to your concerns about the gel image presentation, we have carefully revised the figure to enhance its clarity. The original version showing duplicate lanes has been replaced with a single representative result to eliminate any potential confusion about lane repetition. We have now clearly labeled the major 70kDa band as "GST-ENO1" both on the image itself and in the corresponding figure legend. Additionally, we have specified in the legend that this is a representative result from three independent experiments, along with the full experimental conditions. These modifications have significantly improved the figure's interpretability, and we are grateful for your insightful suggestions that helped strengthen our data presentation.

13、Line 278 Fig 2B – this component of the figure would be better presented as a separate table.

The author’s answer: Thank you for your suggestion regarding Figure 2B. We understand your perspective about presenting the data in table format. However, after careful consideration, we believe the current graphical presentation effectively displays the five potential interacting proteins with their key characteristics (molecular weight, peptide counts, domain information) in a more visually integrated manner that maintains consistency with the overall figure layout. This format allows readers to more intuitively grasp the relationships between these candidates and the subsequent validation experiments shown in other panels. We hope this explanation addresses your concern, but we would be happy to reconsider if you have additional suggestions.

14、Line 343 suggest revision “This research suggests that the accumulation”

The author’s answer: Thank you for your valuable suggestion regarding the wording in Line 343. We fully agree with your perspective on maintaining academic prudence in our conclusions. As you recommended, we have revised the original phrase "This study reveals for the first time" to "This research suggests" to better reflect the inferential nature of our findings. See lines 331 for specific modifications, which have been marked in red.

This modification helps present our discovery of the ENO1-ACTB axis's role in metabolic reprogramming in a more scientifically appropriate manner while preserving the core significance of our findings. We appreciate your insightful comment that has enhanced the precision of our manuscript.

15、Line 423 suggest revision “This research suggests that the accumulation”

The author’s answer: Thank you for your constructive suggestion regarding the wording in Line 423. We have carefully revised the original phrase "These data suggest" to "This research suggests" as you recommended. See lines 410 for specific modifications, which have been marked in red.

16、Line 535 suggest revision "This study has systematically examined the molecular mechanisms by which”

The author’s answer: Thank you for this constructive suggestion. We have added the recommended transitional sentence at the beginning of Line 535 to better connect the mechanistic findings with future research directions. The modification strengthens the logical flow while preserving all proposed research avenues. See lines 523-525 for specific modifications, which have been marked in red.

We sincerely appreciate your thoughtful guidance. We have meticulously revised the paper in accordance with your feedback, aiming to enhance its rigor and accuracy. Should you have any questions or require further clarification, please do not hesitate to reach out to us.

Yours sincerely,

Yong Wang

10, July, 2025

Shihezi University
